# Superior Line from Anther Culture of *Dendrocalamus latiflorus* Selected after Field Trial

**Wei Zhang** [1,†]**, Yujun Wang** [1,2,†]**, Guirong Qiao** [1,2,†]**, Huijin Fan** [1,2]**, Kangming Jin** [1,2]**, Biyun Huang** [1,2]**, Wenmin Qiu** [1,2]**, Yueguo Zou** [3]**, Jinzhong Xie** [1,*] **and Renying Zhuo** [1,2,*]

1  Key Laboratory of Tree Breeding of Zhejiang Province, The Research Institute of Subtropical of Forestry, Chinese Academy of Forestry, Hangzhou 311400, China; zhangwei@caf.ac.cn (W.Z.); yujunwang0618@foxmail.com (Y.W.); gr_q1982@163.com (G.Q.); fhj1201@163.com (H.F.); 17858287379@163.com (K.J.); hby948750582@163.com (B.H.); qiuwm05@163.com (W.Q.)
2  State Key Laboratory of Tree Genetics and Breeding, Chinese Academy of Forestry, Beijing 100091, China
3  Bamboo Garden of Hua'an County, Hua'an 363800, China; fjhalykj@163.com
*  Correspondence: xiejinzhong@caf.ac.cn (J.X.); zhuory@caf.ac.cn (R.Z.); Tel.: +86-571-63311860 (J.X. & R.Z.)
†  These authors contribute equally to this paper.

**Abstract:** The selection of superior lines is extremely important to improve the utilization rate and economic value of bamboo. In this research, 120 anther-regenerated bamboo lines were planted in the field, and the survival rate reached 84.2% one year after planting. During five years of observations, we continuously measured and recorded the number of shoots and the size of the new bamboo of these regenerated lines. The results showed that there were considerable differences in culm size and growth rate among the different lines. After comprehensive evaluation, we found that one of the lines (P82) had obvious advantages in culm size and growth rate compared with the others. The chromosome ploidy of line P82 and the other three lines (P38, P84, and P34) was detected. It was found that P82 was hexaploid, while the other three lines were dodecaploid. Nutritional components of the P82 shoots were further detected. The results showed that the content of soluble sugar was 1.4%, the content of free amino acid was 3.5 g·kg$^{-1}$ (FW, fresh weight), and the content of protein was 14.8 g·kg$^{-1}$ (FW), and there were no significant differences compared with the local wild mature bamboo. Anatomical analysis showed that the vascular bundle size of the line P82 (hexaploid) was significantly larger than that of line P38 (dodecaploid), and the length of parenchyma cells in the culm wall of line P82 was similar to that of line P38, however, the cell width of line P82 was significantly wider than that of line P38. In this study, the breeding of superior lines of regenerated bamboo plants from an anther culture was realized, which provided an example for a new method for selecting superior lines from an anther culture, and also enriched the resources of superior lines of *D. latiflorus*.

**Keywords:** *Dendrocalamus latiflorus*; breeding; anther culture; polyploid

## 1. Introduction

Bamboo is an important forest resource because of its wide distribution, rapid growth, multiple uses, and high ecological and economic value. *Dendrocalamus latiflorus* Munro is the most widely cultivated bamboo species in southern China. It is also one of the clumping bamboo species with the highest utilization rate due to its giant culm and delicious bamboo shoot. Due to its great economic value, people have been trying to cultivate and screen excellent lines in order to further improve its utilization and economic value. Bamboo has special biological characteristics, for example, the sporadically occurring flowering periods and the low rate of natural seed setting, which has largely restricted the development of crossbreeding and the breeding of new varieties [1,2]. Xie et al. [3] used the program colony based on simple sequence repeat (SSR) markers and found that the hybrid affinity of tropical woody bamboo is relatively high. Some studies have observed that the flowering and pollination characteristics and seedling growth characteristics of several sympodial

bamboos made a preliminary selection of seedlings [4–8]. Some researchers carried out the crossing of *Bambusa pervarians × B. glauca*, and excellent hybrids such as *B. pervariabilis × (D. latiflorus + B. textilis)* No. 1, and 7, and *D. latiflorus × D. hamiltonii* No. 1 were selected successfully [9–11]. Ning et al. [12] selected four excellent hybrids, such as No. 3, 6, 8, and 30 by *B. pervariabilis × Dendrocalamopsis daii*. Huang et al. [13] evaluated 10 main bamboo species in China, and 4 excellent bamboo lines were selected with higher nutrition and good flavor of bamboo shoots. Although some excellent lines of *D. latiflorus* have been successfully selected by means of hybridization, it takes decades of hard work. Therefore, it is a time-consuming and difficult way to obtain excellent lines. In this context, anther culture of regenerated lines provides a new method for the efficient breeding of fine lines of *D. latiflorus*.

In the previous work, our research team used immature anthers of *D. latiflorus* as explants to induce callus and differentiate into seedlings, and successfully obtained regenerated lines (these regenerated lines have different ploidy) [14]. In this work, we carried out the field cultivation experiment for the above regenerated lines, and through the observation and evaluation of their growth status, a superior line (P82) was selected. Then, we analyzed the nutrient composition of its bamboo shoots and measured its culm morphology. The results showed that, in terms of culm morphology, P82 had significant advantages compared with other regenerated lines, and its culm diameter was similar to that of the local mature *D. latiflorus*; in terms of food, the protein and free amino acid contents in shoots of the regenerated lines was higher than that of seedling plants.

## 2. Materials and Methods

### 2.1. Experimental Site

The experimental materials of bamboo (*Dendrocalamus latiflorus* Munro) were planted in Hua'an County, Zhangzhou City, Fujian Province, China (117°16′ E, 24°38′ N). It is a transitional area with humid monsoon climate from the south subtropical region to the middle subtropics. The regenerated bamboo forest lies on the gentle slope land, north of the bamboo garden in Hua'an County, with an area of 1 hm$^2$, where the soils were loose brick-red soil (Table 1).

**Table 1.** Conditions of climate, altitude, and soil of the regenerated *D. latiflorus* planting area.

| Climate and Altitude Conditions | Value | Soil Conditions | Value |
| --- | --- | --- | --- |
| Altitude | 155 m | Thickness | >1 m |
| Mean annual temperature | 17.3 °C | pH | 5.32 |
| Maximum temperature | 39.5 °C | Organic matter | 37.89 g·kg$^{-1}$ |
| Minimum temperature | −3.7 °C | Available nitrogen | 233.84 mg·kg$^{-1}$ |
| Mean annual frost-free period | 320 d | Available phosphorus | 50.01 mg·kg$^{-1}$ |
| Mean annual precipitation | 1448–2023 mm | Available potassium | 19.32 mg·kg$^{-1}$ |

### 2.2. Experimental Materials

The regenerated lines involved in this experiment were obtained from bamboo anther culture in 2010, and the detailed works related to anther culture and plant regeneration was published in 2013 [14]. After rooting and transplantation, 120 lines were selected, and each line was considered as a separate genotype, numbered P01 to P120. In order to carry out the control experiment, we germinated the *D. latiflorus* seeds obtained from the flowering bamboo forest with the same origin as the generated plants at the same period of the anther culture experiment. The seedlings obtained were used as the wild type (WT). In addition, for reference, we also measured the relevant data of local wild mature *D. latiflorus* (Local) near the experimental planting site in Hua'an County.

## 2.3. Field Experiment

In March 2011, the regenerated lines and WT plants were planted in the experimental site. Each cluster of bamboo (each line was divided into 3 clusters) was randomly distributed in the same area. The row spacing of each cluster was 3 m × 3 m and the initial planting density was about 1110 clusters·hm$^{-2}$. The same management was implemented, including 1 kg of organic fertilizer (fermented pig manure) that was applied on the bottom of each planting hole before planting, and 0.5 kg of that was applied around the plants in late April every year after planting. Weeding was performed manually from July to August every year. The bamboo clusters were standardized in 2013 (removing the bamboo over 2 years old and leaving 4–6 plants per cluster). For shoot harvesting, the shoots were harvested at ground level with the leaves and shoots' fresh weight (FW) being weighed. Survival rate of regenerated lines in the field was investigated in July 2012, July 2014, and July 2019. The shoot number (SN), ground diameter of new bamboo (GD), and the culm heights of new bamboo (CH) were investigated in July 2012 and July 2014. The FW and SN of some superior *D. latiflorus* lines and WT were determined in July–September 2015. Some superior regenerated lines were selected for comparison the CH and GD with WT and Local in July 2016.

## 2.4. Nutrient Comparison of Bamboo Shoots

Since *D. latiflorus* is one of the most important bamboo species for shoot-use, some nutrient indicators of shoots were tested: the protein content was determined using the semi-trace Kjeldahl method [15]; the soluble total sugar and starch content was determined using the anthrone colorimetric method [16]; the ash content was determined using the muffle furnace burning method [16]; the tannin was determined using the hide powder method [16]; the total free amino acids were colored using the ninhydrin method, and the hydrolyzed amino acids using the liquid chromatography method [17].

## 2.5. Anatomical Comparison of Bamboo Culms

For observing the anatomical characteristic of the bamboo culm among regenerated plants, samples were separated by the 6th (the longest internode) internodes of the culm. These samples were further divided into small pieces with a length of 3 cm and fixed in the FAA solution for 2 days, they were then transferred to the glycerol–ethanol (1:1) solution to soften for at least 15 days. After that, samples were routinely PEG embedded, sliced, and stained, as described in Wang et al. [18]. On the basis of these sections, we measured and counted the vascular bundle size, the diameter of metaxylem vessel, and the parenchyma cell size of culm wall.

## 2.6. Detection of Chromosome Ploidy in Plant Materials

We collected the new leaves of the anther regeneration lines P82, P38, P84, P34, and the local plant of *D. latiflorus* as test materials. We cut up these samples thoroughly in the Otto solution (0.1 mol/L citric acid, 50% Tween-20), then filtered and centrifuged them to make a cell suspension. The genome size of the treated materials was measured by using a FACSCalibur flow cytometer (Becton Dickinson, Franklin Lakes, NJ, USA).

## 2.7. Data Statistics and Analysis

For the growth and shoots' nutrient comparison, WT and Local were used as a reference. The coefficient of variation, which is defined as the ratio of the standard deviation to the average, was used to compare the degree of dispersion between the data groups. The data were analyzed using SPSS 23.0 software. One-way ANOVA was applied to test the statistical significance of nutrition and growth to different lines.

The membership function method was often used to screen superior lines. This method is based on the principle of fuzzy mathematics, using membership function value (MFV) for comprehensive evaluation [19]. The membership function value formula is MFV = $(Xi - Xmin)/(Xmax - Xmin)$, where *Xmax* is the index maximum value, *Xmin* is

the index minimum value, and *Xi* is the measured value of each index. For superior line selection, equal weights were given to the SN, CH, and GD.

## 3. Results

### 3.1. Survival Rate of Regenerated Lines in Field

After being planted in the field, some lines gradually died because of weak growth. By 2012, after one year of planting, 101 regenerated lines remained, accounting for 84.2% of the regenerated lines being planted in 2011. By 2014, after three years of planting, 94 regenerated lines remained in the field. By 2019, 86 regenerated lines remained in the field, accounting for 71.7% of the regenerated lines being planted in 2011.

### 3.2. Difference of Growth Characters in Regenerated Lines after One Year of Planting

The regenerated lines' ability to shoot after one year of planting were different (Table 2). The maximum SN was 12, and the minimum SN was 0, while the mean CH of first shoot was 2.57 times that of the mother bamboo; the maximum CH of the first shoot could reach 184 cm, and the coefficient of variation of CH with first shoot was 1.84 times that of mother bamboo. That indicated there was a great difference in the shooting ability of regenerated lines, which provided the possibility for selecting superior lines. Since the growth status of regenerated lines were still unstable, longer observations were needed to select superior lines.

**Table 2.** Variation of culm heights (CH) and shoot number (SN) in regenerated lines after one year of planting.

|  | Mean | Maximum | Minimum | Maximum Difference | Coefficient of Variation % |
|---|---|---|---|---|---|
| CH of mother regenerated lines (cm) | 19.5 | 35.0 | 3.0 | 32.0 | 28.5 |
| CH of first shoot (cm) | 50.1 | 184.0 | 12.0 | 172.0 | 52.4 |
| SN per cluster | 2.9 | 12.0 | 0 | 12.0 | 45.1 |

### 3.3. Differences in Growth Characters of Regenerated Lines after three Years of Growth and Selection for Superior Regenerated Lines

After three years being planted in the field (2014), there was a large range of variation among the regenerated lines in terms of SN (Figure 1a), CH (Figure 1b), and GD (Figure 1c). In 2014, the coefficients variation of SN, GD, and CH reached 51.4%, 41.4%, and 39.3%, respectively. The CH and GD of regenerated line P82 were higher than those of other regenerated lines (Table 3). The relevant values of all 94 surviving lines were shown in Table S1.

Superior line was selected by the highest MFV (Table 3). The best line (P82) was selected for future study (shoots' yield, ploidy level, shoots' nutrient, and culm anatomical properties).

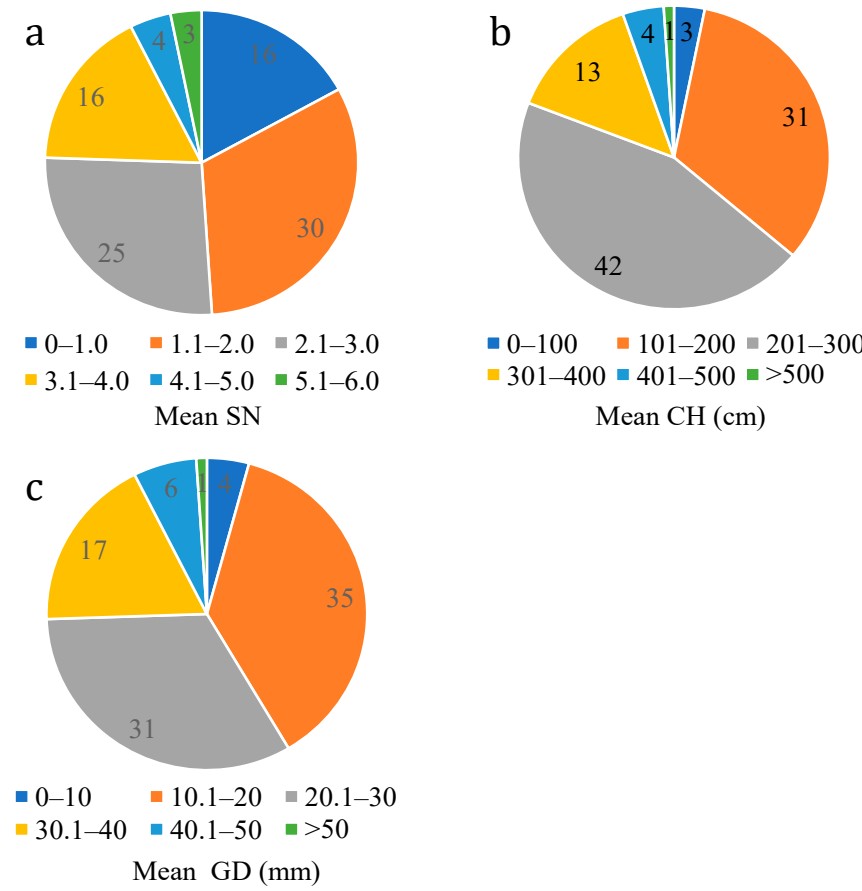

**Figure 1.** Distribution of mean shoot number (SN) (**a**), mean culm heights (CH) (**b**), and mean ground diameter (GD) (**c**) to the number of regenerated lines after three years of planting.

**Table 3.** Mean and MFV of shoot number (SN), culm heights (CH), and ground diameter (GD) of top 20 regenerated lines after three years of planting.

| Plant Label | SN | | CH (cm) | | GD (mm) | | Mean MFV |
|---|---|---|---|---|---|---|---|
| | Mean ± SD | MFV | Mean ± SD | MFV | Mean ± SD | MFV | |
| P82 | 3.7 ± 0.5 | 0.576 | 697.3 ± 87.2 | 1.000 | 61.9 ± 6.8 | 1.000 | 0.859 |
| P84 | 2.5 ± 1.5 | 0.364 | 420.3 ± 100.1 | 0.614 | 40.5 ± 12.8 | 0.571 | 0.516 |
| P38 | 4.7 ± 0.9 | 0.758 | 269.7 ± 16.2 | 0.416 | 29.5 ± 5.5 | 0.338 | 0.504 |
| P34 | 5.0 ± 0.0 | 0.818 | 260.0 ± 39.6 | 0.304 | 23.4 ± 5.8 | 0.322 | 0.482 |
| P03 | 3.0 ± 1.0 | 0.455 | 379.7 ± 9.9 | 0.445 | 31.2 ± 0.8 | 0.508 | 0.469 |
| P20 | 1.0 ± 0.1 | 0.091 | 459.7 ± 106.7 | 0.681 | 44.2 ± 6.3 | 0.633 | 0.468 |
| P18 | 3.3 ± 1.9 | 0.515 | 314.7 ± 18.4 | 0.458 | 31.9 ± 1.9 | 0.408 | 0.460 |
| P109 | 2.5 ± 0.5 | 0.364 | 335.3 ± 34.5 | 0.563 | 37.7 ± 4.0 | 0.439 | 0.455 |
| P37 | 3.0 ± 1.6 | 0.455 | 330.3 ± 36.3 | 0.477 | 32.9 ± 5.6 | 0.431 | 0.454 |
| P116 | 1.0 ± 0.2 | 0.091 | 450.0 ± 39.8 | 0.652 | 42.6 ± 4.6 | 0.617 | 0.453 |
| P22 | 1.0 ± 0.1 | 0.091 | 375.0 ± 144.7 | 0.762 | 48.7 ± 20.1 | 0.501 | 0.451 |
| P40 | 3.7 ± 0.5 | 0.576 | 264.7 ± 44.1 | 0.429 | 30.3 ± 6.1 | 0.330 | 0.445 |
| P45 | 2.3 ± 0.9 | 0.333 | 368.7 ± 121.2 | 0.493 | 33.8 ± 9.8 | 0.490 | 0.439 |
| P33 | 1.7 ± 0.5 | 0.212 | 378.0 ± 70.4 | 0.586 | 39.0 ± 5.8 | 0.506 | 0.435 |
| P16 | 1.0 ± 0.1 | 0.091 | 403.0 ± 73.4 | 0.640 | 41.9 ± 6.8 | 0.543 | 0.425 |
| P56 | 4.5 ± 0.5 | 0.727 | 235.3 ± 14.8 | 0.229 | 19.2 ± 3.0 | 0.283 | 0.413 |
| P14 | 1.5 ± 0.5 | 0.182 | 329.7 ± 10.2 | 0.618 | 40.8 ± 0.9 | 0.431 | 0.410 |
| P69 | 4.0 ± 0.0 | 0.636 | 225.0 ± 54.9 | 0.308 | 23.6 ± 9.6 | 0.268 | 0.404 |
| P28 | 3.0 ± 0.0 | 0.455 | 310.3 ± 121.6 | 0.352 | 26.0 ± 6.5 | 0.400 | 0.402 |
| P94 | 2.5 ± 0.5 | 0.364 | 275.3 ± 5.2 | 0.495 | 33.9 ± 1.4 | 0.345 | 0.401 |

Note: mean ± standard deviation (SD) (n = 3), membership function value (MFV) was described in 2.7 Data Statistics and Analysis.

### 3.4. Comparison the Shoot' Yield of Superior Lines

The yield of top three regenerated lines (P82, P84, P38) were determined in 2015. Although the SN of P82 was not significantly different ($p < 0.05$) to that of P38 and WT, the FW of P82 was significantly higher ($p < 0.05$) than that of P84, P38, and WT (Figure 2).

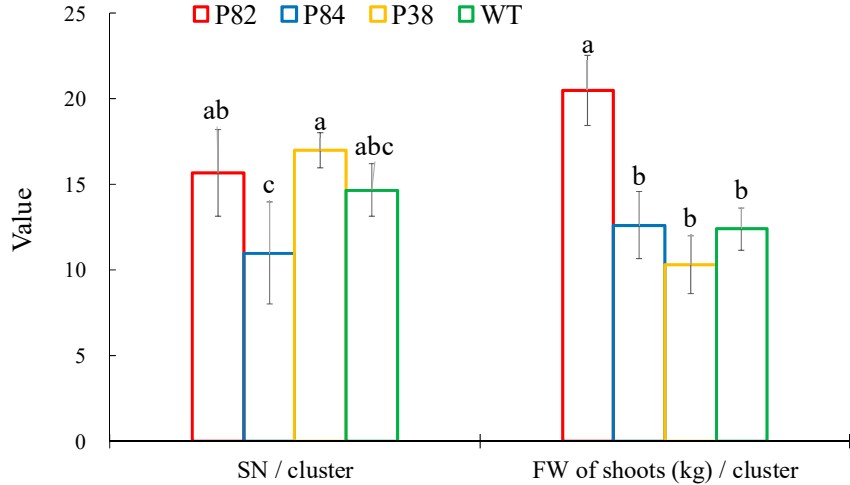

**Figure 2.** Comparison of the shoot number (SN) and fresh weight (FW) to some *D. latiflorus* lines. Mean $\pm$ SD (n = 3), different letters indicate significant differences at $p < 0.05$ level.

Through five years of observation, the new culm of line P82 reached the CH (Figure 3a) and GD (Figure 3b) of the Local faster than those of P84, P38, and WT.

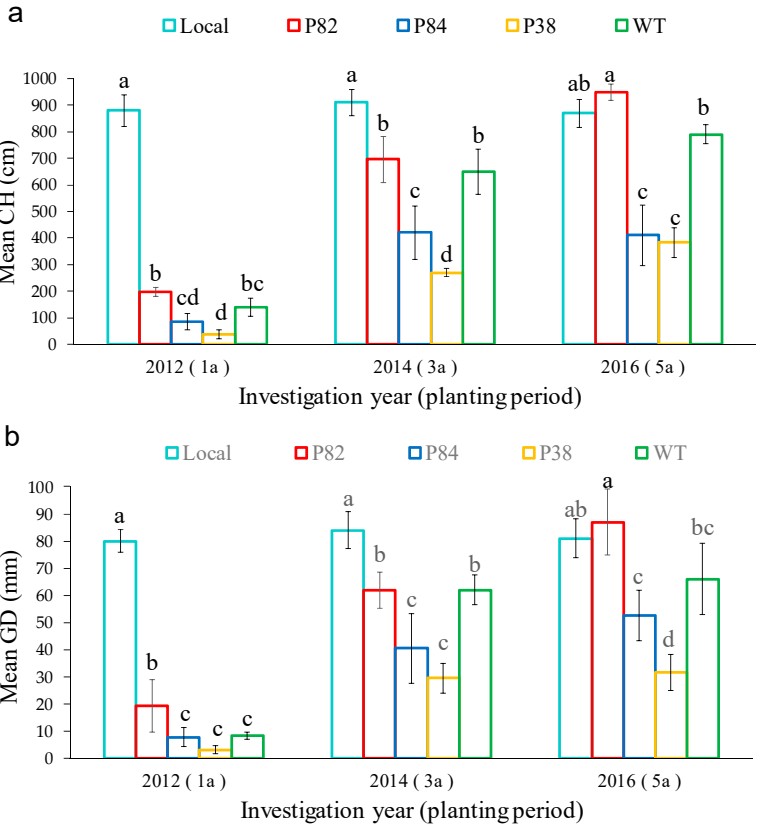

**Figure 3.** Comparison of mean culm heights (CH) (**a**) and ground diameter (GD) (**b**) to new culm of some *D. latiflorus* lines. (1a), (3a), and (5a) represent the year after cultivation, respectively. Mean $\pm$ SD (n = 3). Different letters indicate significant differences at $p < 0.05$ level.

### 3.5. Ploidy Level Test of Superior Lines P82

The DNA content of P82, P38, P84, P34, and the Local (the Local group mentioned in this article belongs to this type) were detected by flow cytometry to determine the chromosome ploidy (Figure 4). The Local has been proven to be naturally hexaploid [20]. Compared with its DNA content, the result shows that the superior line P82 was also hexaploid (Figure 4a), while the lines P38, P84, and P34 were all dodecaploid (Figure 4b).

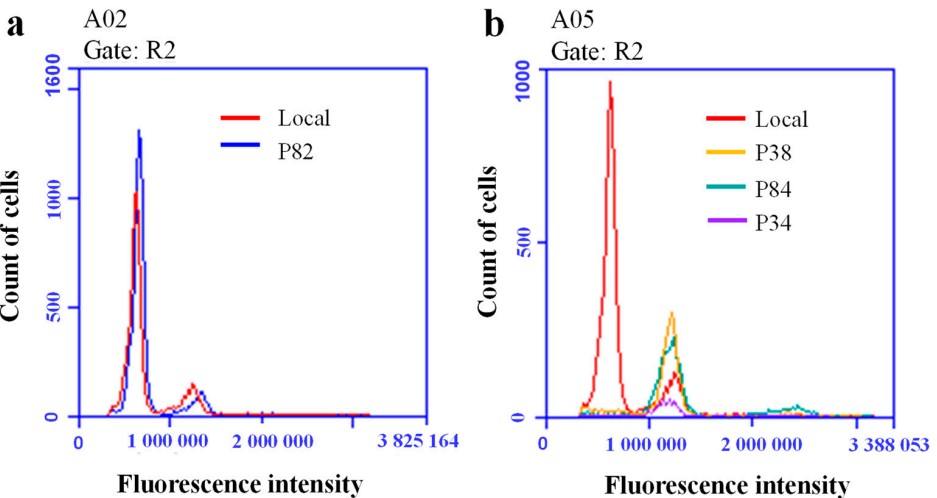

**Figure 4.** Comparison of DNA content in the different lines of *D. latiflorus*. Fluorescence intensity was indicative of DNA content. (**a**) DNA content of line P82 and Local (It is known to be hexaploid); (**b**) DNA content of lines P30, P84, P34 and Local.

### 3.6. Comparison the Shoot' Nutrient of Superior Lines

The results of the shoots' nutrients showed that (Table 4), in general, the mean contents of proteins, free amino acids, and hydrolyzed amino acids in regenerated lines (P38, P84, P82) were significantly higher ($p < 0.05$) than those of WT.

**Table 4.** Shoots' nutrient of some *D. latiflorus* lines.

| Plant Label | Ash (g·100 g$^{-1}$) | Soluble Sugar (%) | Free Amino Acid (g·kg$^{-1}$ FW) | Protein (g·kg$^{-1}$ FW) | Hydrolyzed Amino Acids (g·kg$^{-1}$ FW) | Tannin (g·kg$^{-1}$ FW) | Oxalic Acid (g·kg$^{-1}$ FW) |
|---|---|---|---|---|---|---|---|
| P38 | 0.8 ± 0.08 a | 1.5 ± 0.3 a | 3.2 ± 0.2 a | 16.4 ± 2.8 a | 9.4 ± 2.1 a | 2.5 ± 0.2 a | 1.8 ± 0.1 a |
| P84 | 0.6 ± 0.07 b | 1.0 ± 0.1 b | 3.6 ± 0.3 a | 17.0 ± 2.3 a | 10.4 ± 2.3 a | 2.2 ± 0.4 ab | 2.1 ± 0.3 a |
| P82 | 0.6 ± 0.05 b | 1.4 ± 0.1 ab | 3.5 ± 0.5 a | 14.8 ± 1.4 a | 9.1 ± 2.3 a | 2.0 ± 0.4 ab | 1.8 ± 0.2 a |
| WT | 0.5 ± 0.03 b | 1.1 ± 0.1 b | 1.8 ± 0.1 b | 9.6 ± 1.5 b | 6.4 ± 0.6 b | 1.6 ± 0.2 b | 2.1 ± 0.3 a |
| Local | 0.6 ± 0.04 b | 1.2 ± 0.3 ab | 3.5 ± 0.5 a | 17.2 ± 3.4 a | 11.9 ± 1.7 a | 2.4 ± 0.5 ab | 1.0 ± 0.1 b |

Note: Mean ± SD (n = 3), different letters above indicate significant differences between different lines (n = 3, $p < 0.05$).

### 3.7. Culm Anatomical Properties of Superior Lines

In all anther regenerated lines, we detected two different chromosome ploidies, the hexaploid line (P82) and the dodecaploid lines (P38, P84, etc.), as shown in Figure 4. The culm size showed great differences between the two different ploidy plants, as shown in Figure 3. In order to compare the morphology of culm wall tissue between them, we investigated the anatomical properties of the culms of the P82 line and the P38 line. Results showed that the morphology of the vascular bundles was consistent in the two lines (Figure 5a,b), but the width and length of the vascular bundles in P82 were significantly bigger ($p < 0.01$) than those of the dodecaploid lines (Figure 5c). Interestingly, no significant difference ($p < 0.01$) was observed for the diameter of the metaxylem vessels, which were the main channel of water transport between them (Figure 5c). The longitudinal section

in the outer layer of the culm wall showed line P82 had more cellular layers in the cortex position than line P38 (Figure 5d,e). In addition, higher lignification in the parenchyma cells could be seen in line P82, whose cell wall was stained red by Safranin O, compared with line P38 (Figure 5f,g). Size comparison of the parenchyma cells showed that the cell wide (CW) of line P82 was significantly larger ($p < 0.01$) than that of line P38. However, no significant differences ($p < 0.01$) were observed for the cell length (CL) or the ratio of cell wide/cell length (CL/CW) (Figure 5h).

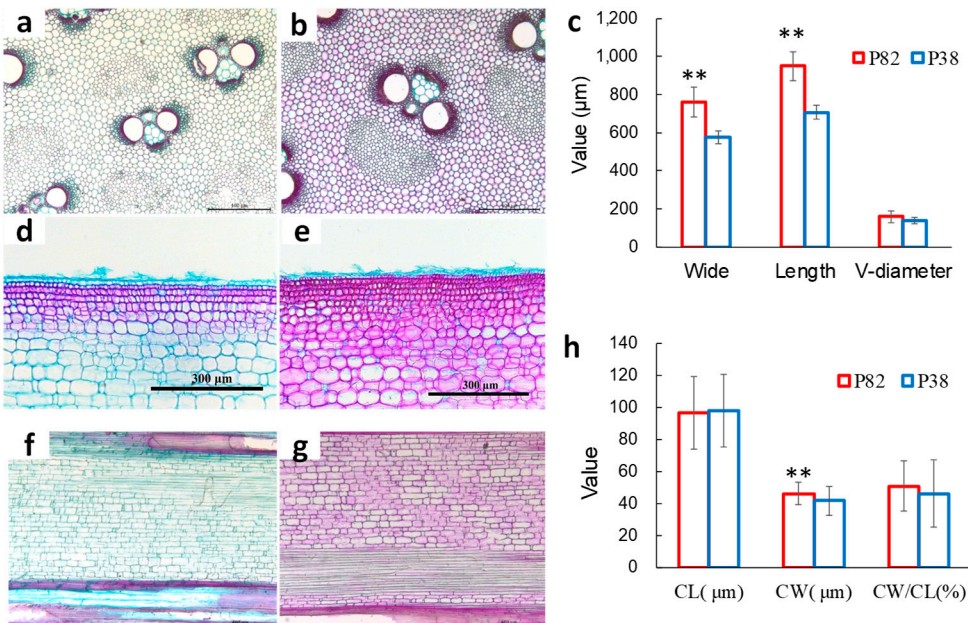

**Figure 5.** Cytological analysis of the culm of line P82 and line P38. (**a**,**b**) Cross-sections of the culm of line P38 and line P82, respectively; (**c**) comparative analysis of the vascular bundle size and diameter of the metaxylem vessel in line P82 and line P38; (**d**,**f**) longitudinal section of the culm in the outer layer and inner part of line P38; (**e**,**g**) longitudinal section of the culm in the outer layer and inner part of line P38. (**h**) Comparative analysis of the parenchyma cell size in line P82 and line P38. CL, cell length; CW, cell wide; CW/CL (%), cell wide/cell length (%). ** Indicate significant differences at $p < 0.01$ level.

## 4. Discussion

The flowering times of bamboo are uncertain, which has largely restricted the development of crossbreeding and the breeding of new varieties [21]. Under the circumstances, asexual propagation, such as cutting, has long been adopted in the breeding of bamboo plants; however, it also will bring a series of problems, such as bamboo forest aging. In view of this dilemma, the method of anther culture can effectively avoid the abovementioned problems. Due to the recombination of genes in the process of meiosis, some recessive genes regulating important traits may exist in the germ cells of anthers alone; however, plants regenerated from these cells can provide a basis for the selection of superior lines. Of course, an anther culture will also lead to chromosomal diversity in the regenerating process due to the fact that some regenerated lines will be differentiated from somatic cells, while some others will be differentiated from germ cells. Besides, chromosome doubling also occurs easily during tissue culture from haploid cells and tissues [14,22], this also enriches the gene pattern of the regenerated lines and is conducive to the selection of superior lines [23].

In this study, 120 regenerated lines from an anther culture were cultivated in the field in 2011. In the three years after transplanting, we observed that the survival rate of the regenerated lines was higher (the survival rate was more than 78%). In addition, we also found that there were great differences in culm size and SN among the different

regeneration lines (Table 2, Figure 1). When the survey was performed in August 2012, with the exception of P82, the mean SN, CH, and GD of the regenerated lines were lower than those of WT seedlings, which have same age as the regenerated lines (Figure 2). The results of continuous observation showed that the culm diameter of P82 was significantly higher than that of WT and other regenerated lines of the same age in the first year of planting. Additionally, in the fifth year of planting, the GD and CH of P82 were significantly higher than those of the same age WT and other regeneration lines, and even higher than those of mature local *D. latiflorus* (Figure 3). This result indicated that the line P82 had an extraordinary growth rate.

Yang et al. [7] observed the seedling growth of the hybrid offspring of *D. latiflorus* at the seedling stage and found that the correlation coefficients between SN, GD, and CH did not reach a significant level. These results suggested that shooting ability cannot be selected only by SN, GD, or CH. As *D. latiflorus* is a dual-purpose bamboo species used for shoots and wood, the SN and biological growth should be given a certain weight to select excellent lines for different applications. In this study, the weight of the three characteristics (SN, CH, and GD) were also based on the above consideration. After the comprehensive evaluation of the above factors, we selected P82 as the superior line in this batch of anther regenerated lines (Table 3).

In previous studies, we know that 96% of these regenerated lines are dodecaploid [14], and its wild type was confirmed to be hexaploid. Interestingly, we found that P82 was a hexaploid line (Figure 4). Given the short stature of all other dodecaploid regenerated lines, this result may mean that the optimal chromosome ploidy of *D. latiflorus* is hexaploid. Polyploidy breeding, as an important method, has been achieved in species such as barley, wheat, rice, maize, tobacco, sugar beet, and onion [24,25]. Autoploid plants always exhibit phenotype with cell size and number increases, especially in meristematic tissues. Polyploid plants usually have thicker, broader, and shorter leaves. Other plant organs may increase in size compared to their corresponding parts in diploids, an effect called gigas features [26]. However, this obviously does not apply to *D. latiflorus*.

In this experiment, the basic nutritional composition of the bamboo shoots showed differences in the different lines. P82 had the potential for the best bamboo shoot nutrients of the lines studied (Table 4). The quality of bamboo shoots is mainly affected by genetic factors [27]. In addition, nutrients in the soil and the management also affect the nutrients in bamboo shoots. However, because intensive management measures were not adopted in this experiment and the bamboo shoots of some lines were relatively small, the nutrients of bamboo shoots measured may be different from those of adult bamboo shoots planted in fertile soil with high-intensity management measures, so the further analysis is needed.

Due to the significant difference in culm size between P82 and the same batch of regenerated dodecaploid lines, we compared the anatomical structure of the culm of these two lines. The results showed that the vascular bundle size in P82 was significantly larger than that of line P38 (dodecaploid). In addition, the results of Safranin O staining showed that P82 appeared to have higher lignification degree (Figure 5). These results show that the culm wall of P82 has strong mechanical support, which supports its great development potential in size to a certain extent. Besides, the length of the parenchyma cells in the culm wall of P82 was similar to line P38; however, the cell width of P82 was significantly wider than P38 (Figure 5), which meant that P82 had larger cell sizes in parenchymal tissue. The parenchyma cells in the culm wall are the main places for the accumulation of proteins and other nutrients [28], which means that P82 has stronger nutrient storage capacity.

## 5. Conclusions

In summary, through a field cultivation experiment, we found that the regenerated lines of *D. latiflorus* by anther culture had a good survival rate, but obvious differences in the growth state among the different lines were also present. After continuous monitoring, the superior line P82 was selected, with significant dominant characteristics from 120 regenerated lines. This superior line not only has better improvement in the nutrients

of its shoots than other regeneration lines and even the local wild *D. latiflorus*, but also has extremely fast growth speed and greater timber value. Our work can not only enrich the superior line resources of *D. latiflorus*, but also contribute to the improvement of its utilization efficiency.

**Supplementary Materials:** The following are available online at https://www.mdpi.com/article/10.3390/horticulturae7050098/s1, Table S1: Mean and MFV of shoot number (SN), culm heights (CH), and ground diameter (GD) of regenerated lines after three years of planting.

**Author Contributions:** R.Z. and J.X. conceived and designed the experiments. W.Z., Y.W., G.Q., and Y.Z. performed the experiments. H.F., K.J., B.H., and W.Q. analyzed the data. W.Z. and Y.W. wrote the manuscript. All authors have read and agreed to the published version of the manuscript.

**Funding:** This work was supported by Funding for this study was provided by National Key Research and Development Program of China (No.2016YFD0600903) and Basic Public Welfare Research Program of Zhejiang Province (No. LGN18C160007).

**Institutional Review Board Statement:** Not applicable.

**Informed Consent Statement:** Not applicable.

**Data Availability Statement:** Data is contained within the article or supplementary material.

**Conflicts of Interest:** The authors declare no conflict of interest.

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
