# Peer review of "Superior Line from Anther Culture of Dendrocalamus latiflorus Selected after Field Trial"

_horticulturae, doi:10.3390/horticulturae7050098_

Round 1
Reviewer 1 Report
Pollen embryogenesis is one of the main methods of plant breeding. It is also applied in the grass family. The plants regenerated from immature microspores are characterised by various genotypes resulting from meiotic recombination. Research on the morphological and genetic diversity of doubled haploids obtained in tissue cultures of different species is widely conducted. Due to the cultural and economic importance of Dendrocalamus latiflorus I find the performed analysis useful. However, I have a number of remarks about the layout of the experiment, description of the results and the quality of the discussion and conclusion.
Materials and Methods Section:
The authors did not describe the plants studied in the experiment, e.g. what kind of cross the F1 hybrid comes from, or what variety of “Local D. latiflorus” was used.
L. 79-83: Is the macronutrient content of the soil the same in the entire experimental field?
L. 84-88: How has plant ploidy been estimated? Can the ploidy of dodecaploid plants result from regeneration from somatic cells and the subsequent multiplication of the number of chromosomes through the use of mutagens? How were the tested plants obtained?
L. 97-101: In my opinion, comparing 1-year-old and 2-year-old plants in terms of morphological features is a methodological error.
L. 104-109: Please describe the methods used in detail or quote literature supporting them.
L. 111-115: The sentence is not clear.
L. 117-120: How were cytological specimens prepared?
L. 121-125: Control group should not be the average value of the observations (all clones). In the further part of the experiment, control groups were hybrid F1 and “Local D. latiflorus” – there is no information about these plants.
Results Section:
Figure 1 - Description and formatting should be corrected.
L. 156-159: This information is an exact repetition of the results presented in table 2.
L. 189-190: Value of average contents of ash and soluble sugar does not match the data from table 4.
Table 4: Please improve formatting. Why are in this part of the experiment control groups F1 hybrid and “Local D. latiflorus”?
Figure 3c: Figure illegible.
Figure 4: Improve formatting.
Discussion Section:
L. 235-237: “The genetic variation types of regenerated plants from anther culture were more abundant than those from conventional hybrids.” - I do not agree with this statement.
L. 239-242: Anther culture has not been successful in soybean yet.
L. 244-247: Sentence is not clear.
L. 250-253: What variety of bamboo was described here?
L. 262-264: It is obvious that homozygous genes caused by chromosome duplication lead to loss of heterosis.
L. 283-289: Does it mean that androgenesis is not the right method in bamboo breeding?
Conclusions Section:
There is no conclusion; rather summary. This chapter should be completely rewritten.
L. 292-307: This information is inadequate for this section.
L. 302-307: Increasing the number of chromosomes so that they exceed critical values for a given species is not preferred. This phenomenon is described by many researchers.
L. 311-312: “P82 had better eating quality 311 (higher soluble sugar content and lower tannin content) than local D. latiflorus.” - this sentence is inconsistent with the presented results - the presented differences are not statistically significant.
To sum up - the work can only be published after major corrections have been made.
Reviewer 2 Report
Dear authors,
please consider changing average value for mean value. It is also necessary to modify description below the horizontal axis in Figure 1.
Author Response
Dear reviewers:
Thank you for your time and effort in handing our manuscript titled “Superior clones from anther culture of Dendrocalamus latiflorus selected after field trial”. We have revised our manuscript in light of the valuable suggestions of the reviewers. Below you will find our detailed responses to the specific comments of the reviewers. Comments are pasted below in italics and our responses are in regular blue text.
1. please consider changing average value for mean value. It is also necessary to modify description below the horizontal axis in Figure 1.
Responses:Thank you for pointing this out. We have improved all the figure in our manuscript according to your suggestion.
Reviewer 3 Report
It is a very interesting study. Please see the notes inserted in the text

Reviewer 4 Report
- The authors used anther culture to regenerate plants of different ploidy plants, however, the methodology is not described, only one citation of a brief report with no ploidy analysis or confirmation of the tissue origin of the regenerated plants.
- In 2.1 Experimental materials, I wonder how the annual frost-free period could be 5320 days? The maximum days is 365 days. Or the data was accumulated many years?
- Authors claim two ploidy types were observed, but no evidence to support. Also if the dodecaploid plants were indeed derived from microspores, the growth rate of these plants may be slower than the regular hexaploid. But no data showing this point.
Author Response
Dear reviewers:
Thank you for your time and effort in handing our manuscript titled “Superior clones from anther culture of Dendrocalamus latiflorus selected after field trial”. We have revised our manuscript in light of the valuable suggestions of the reviewers. Below you will find our detailed responses to the specific comments of the reviewers. Comments are pasted below in italics and our responses are in regular blue text.
Sincerely ,
Wei Zhang,
Email: jadezh@vip.163.com
Yujun Wang
Email: yujunwang0618@foxmail.com
Reviewer 4
1.The authors used anther culture to regenerate plants of different ploidy plants, however, the methodology is not described, only one citation of a brief report with no ploidy analysis or confirmation of the tissue origin of the regenerated plants.
Responses:Thank you for pointing this out. Anther culture and plantlet regeneration is a previous work of our research team, and the detailed method and process have been published in 2013. As for the chromosome ploidy of regenerated plants, we have supplemented the relevant flow cytometry test data in the manuscript to provide evidence for the confirmation of plant ploidy.
- In 2.1 Experimental materials, I wonder how the annual frost-free period could be 5320 days? The maximum days is 365 days. Or the data was accumulated many years?
Responses:Thank you for pointing this out. This is indeed a mistake and we are sorry for it. We have corrected it in the article.
- Authors claim two ploidy types were observed, but no evidence to support. Also if the dodecaploid plants were indeed derived from microspores, the growth rate of these plants may be slower than the regular hexaploid. But no data showing this point.
Responses:Thank you for pointing this out. We have supplemented the relevant data for the detection of plant ploidy. Indeed, we did not do more experiments to find out the reason for the slow growth of Dodecaploid in this experiment, which may be an interesting topic.
Round 2
Reviewer 1 Report
The manuscript has been largely improved, but still requires some corrections and explanations:
I get the impression that the use of 2 controls in the experiment: WT and Local, is confusing. Authors sometimes use the comparison to one, another time to the other control, and sometimes to both. Moreover, in Conclusions they compare the P82 line only to the local variety (Local).
127-128: Why was colchicine used before measuring the level of ploidy?
95-100 and 117-118: Check the years of observation. Give the abbreviated names of the tested parameters when they are first described in the text.
103-108: There is still no literature describing all the methods used.
Figure 1: Per plant or per line? Titles under individual graphs is not necessary - it is included in the description of the figure.
204: Why the term "probably" appeared. Are the authors unsure?
Figure 5: Does IN/10 mean internode number per 10 plants? - no explanation in the description.
Figure 5a: The figure is missing Local.
Figure 5b: The term “species” is inadequate and written with small letter. P82 is the line. The description of the figure does not mention Dodecaploid.
Figure 6: 246-250: Correct the description.
Conclusion: “Strain” is not valid for a line P82.
Table S1-S2: The first letter of the header should be capitalized.
Table S3: Correct editing, photos are of different size.
Reviewer 3 Report
the work has improved a lot. Some points remain to be improved, mainly with regard to the English construction of sentences,
The abstract needs to be improved and focused more on the research presented
Plaese read the Cover letter

Reviewer 4 Report
- The flow cytometric data are presented that can support the ploidy of regenerated lines , but your conclusion based on Fig. 4 and the lines 204-205: "while the P38, P84 and P34 were all Dodecaploid" may mislead readers or misinterpret the data since you only have this limited evidence. I would suggest to use the term and example as: while the P38, P84 and P34 were probably all dodecaploid".
- I don't prefer to use "strains" in the title for plants which are usually not routinely used term. Line or lines probably are more suitable term in this case and generally used in horticultural science.
Author Response
Dear reviewer:
Thank you for your time and effort in handing our manuscript titled “Superior lines from anther culture of Dendrocalamus latiflorus selected after field trial”. We have revised our manuscript in light of the valuable suggestions of you. Below you will find our detailed responses to the specific comments of the reviewers. Comments are pasted below in italics and our responses are in regular blue text.
We hope that we have addressed the reviewers’ concerns and that you please consider this revised manuscript for publication in Horticulturae.
Sincerely ,
Wei Zhang,
Email: jadezh@vip.163.com
Yujun Wang
Email: yujunwang0618@foxmail.com
Responses to referees
- The flow cytometric data are presented that can support the ploidy of regenerated lines , but your conclusion based on Fig. 4 and the lines 204-205: "while the P38, P84 and P34 were all Dodecaploid" may mislead readers or misinterpret the data since you only have this limited evidence. I would suggest to use the term and example as: while the P38, P84 and P34 were probably all dodecaploid".
Responses: Thanks for this suggestion. We have made corresponding changes according to the suggestion.
- I don't prefer to use "strains" in the title for plants which are usually not routinely used term. Line or lines probably are more suitable term in this case and generally used in horticultural science.
Responses:Thanks for pointing this out. In fact, we are also hesitant about the word “strain”. We have changed “strains” to “lines”. Thank you again for your suggestion.